# Chiroptical activity of hydroxycarboxylic acids with implications for the origin of biological homochirality

Jana Bocková [1✉], Nykola C. Jones [2], Uwe J. Meierhenrich[1], Søren V. Hoffmann [2] & Cornelia Meinert [1✉]

Circularly polarised light (CPL) interacting with interstellar organic molecules might have imparted chiral bias and hence preluded prebiotic evolution of biomolecular homochirality. The L-enrichment of extra-terrestrial amino acids in meteorites, as opposed to no detectable excess in monocarboxylic acids and amines, has previously been attributed to their intrinsic interaction with stellar CPL revealed by substantial differences in their chiroptical signals. Recent analyses of meteoritic hydroxycarboxylic acids (HCAs) – potential co-building blocks of ancestral proto-peptides – indicated a chiral bias toward the L-enantiomer of lactic acid. Here we report on novel anisotropy spectra of several HCAs using a synchrotron radiation electronic circular dichroism spectrophotometer to support the re-evaluation of chiral bio-markers of extra-terrestrial origin in the context of absolute photochirogenesis. We found that irradiation by CPL which would yield L-excess in amino acids would also yield L-excess in aliphatic chain HCAs, including lactic acid and mandelic acid, in the examined conditions. Only tartaric acid would show "unnatural" D-enrichment, which makes it a suitable target compound for further assessing the relevance of the CPL scenario.

[1] University Côte d'Azur, CNRS, Institute de Chimie de Nice, UMR 7272, Nice, France. [2] ISA, Department of Physics and Astronomy, Aarhus University, Aarhus C, Denmark. ✉email: jana.bockova@univ-cotedazur.fr; cornelia.meinert@univ-cotedazur.fr

Enantiomerically pure α-hydroxycarboxylic acids (HCAs) are naturally occurring compounds involved in various metabolic pathways and cellular processes. Moreover, they play a major role as chiral auxiliaries for numerous synthetic applications[1,2]. Despite the potential role of HCAs in the compartmentalisation during the emergence of life[3], hydroxy acids have only recently gained increasing interest as monomeric building blocks of ancestral proto-peptides with heterogeneous backbone architectures, so-called depsipeptides[4–6]. Though the appearance of chiral HCAs in the prebiotic chemistry pool can either be understood by endogenous[7–10] and/or exogenous sources[11,12]; studies on their stereochemical preference are rather scarce. For example, lactic acid is found in Earth's biosphere in both stereoisomeric forms, however, is prevalent in its L-form in plants and higher-level organisms[13–15]. On the other hand, only L-enantiomers of malic and tartaric acids—two hydroxy dicarboxylic acids—occur naturally.

How biomolecular homochirality, assumed to be essential for life[16–20], emerged from an environment of equal amounts of L- and D-stereoisomers is still critically debated[21]. The presence of enantiomerically enriched L-amino[21,22] and D-sugar acids[21,23] in meteorites supports the hypothesis of an extra-terrestrial chiral force responsible for the initial symmetry breaking[24,25]. Stellar ultraviolet circularly polarised light (UV CPL) is often recognised as the most plausible cause for the detection of extra-terrestrial amino acids with large enantiomeric excesses (ee-s) of the same handedness as terrestrial proteinogenic amino acids. Several experimental studies have already shown that monochromatic UV CPL is capable of inducing chiral bias in amino acids by preferential destruction of one enantiomer over the other[26–28] or preferential photosynthesis[29,30]. Moreover, infrared CPL has been detected in the Orion (degree of circular polarisation 17%)[31] and NGC 6334 V (degree of circular polarisation 22%)[32] star-forming regions and it is considered to extend into the UV spectral region, the detection of which is hampered by extensive extinction by dust particles[33].

In contrast to amino and sugar acids[21], the enantioselective analyses of meteoritic samples have not revealed any detectable ee of monocarboxylic acids (MCA)[34] and amines[35]. Our earlier electronic circular dichroism (ECD) and anisotropy spectroscopy experiments on selected amino acids, MCAs and amines in aqueous solution[36] confirmed that this is in agreement with the CPL hypothesis, since they showed about an order of magnitude lower values of the anisotropy factor $g$ compared with amino acids. The anisotropy factor $g$ is given by the ratio between the ECD signal and the absorbance such that $g = \frac{\Delta \varepsilon}{\varepsilon}$, where $\Delta \varepsilon = \varepsilon_L - \varepsilon_R$; $\varepsilon_L$ and $\varepsilon_R$ are extinction coefficients of left-CPL and right-CPL, respectively; and $\varepsilon = \frac{\varepsilon_L + \varepsilon_R}{2}$ is the extinction coefficient. The anisotropy factor $g$ is directly related to the photochemically inducible $ee$[37] and the lower limit can be approximated as follows[38]

$$|\%ee| \geq \left(1 - (1-\xi)^{\frac{|g|}{2}}\right) \times 100\% \qquad (1)$$

where $\xi$ is the extent of reaction. The sign of $ee$ from relation (1) is then determined based on the sign of $g$ at a particular wavelength and helicity of CPL. Anisotropy spectroscopy provides therefore direct information on the polarization- and wavelength-dependent molecular anisotropy $g$ inherent to CPL-induced photochemical processes, as well as the potential outcome in terms of inducible optical purity ($ee$).

Up till now, HCAs have not been reported to be present in a detectable $ee$ in carbonaceous chondrites, except for lactic acid with an $\%ee_L$ of 3–12% in Murchison, GRA 95229 and LAP 02342 meteorites[39]. In general, the enantioselective analyses of HCAs in meteorites are often accompanied by poor enantio-separation, peak coelution, quantification uncertainties of more than 5%[11,39–42] or detection limits are not stated[23], so that reported $\%ee_L$ values have to be handled critically. So far, no experimental study has thoroughly focused on elucidating the $ee$ inducible in lactic acid compared with the other HCAs, and whether the natural stereochemistry of amino and hydroxy acids may share a common heritage.

Although ECD spectra of lactic acid and several other HCAs in aqueous solution have been investigated experimentally and/or theoretically[43–52], this paper is to our knowledge the first reporting the UV-ECD of both D- and L-enantiomers of lactic, 2-hydroxybutanoic, 2-hydroxy-3-methylbutanoic, 2-hydroxy-4-methylpentanoic, malic, and tartaric acids along with their anisotropy spectra and discussion of their potential implications for the generation of prebiotic chiral bias. It is well-known that the surrounding environment of chiral species can significantly affect their chiroptical response[53]. However, the exact environmental conditions and conformations of the molecules in the interstellar environment which experience interaction with CPL during the Solar System formation are not known. Water-dominated interstellar ices represent the most abundant solid-phase components of dense molecular clouds and of the outer part of protoplanetary disks. Photons with energies of 5–9 eV (~248–138 nm) have mean free paths comparable to the thickness of interstellar ices (~0.01 μm thick), which suggests that the majority of the ice layers covering interstellar dust particles would evolve through photochemical reactions[54]. Undoubtedly, interstellar ices which are subject to energetic processing are a rich source of complex organic compounds[55]. The formation of HCAs has been confirmed in several experiments simulating the production of organic species under astrophysical conditions, i.e., interstellar/cometary ice analogues at temperatures <80 K exposed to UV photons/energetic particles[12,56,57]. Interestingly, Tachibana et al.[58] showed that UV-irradiated interstellar ice analogues composed of water, methanol and ammonia, as well as of pure water show liquid-like behaviour over 65–150 and 50–140 K temperature ranges, respectively, which may enhance the formation of complex organic species. Such ices, which are supposed to contribute to the organic inventory of meteorites, micrometeorites and interplanetary dust particles, could have served as a key source of biomolecular precursors for the prebiotic chemical origins of life[59,60]. Therefore, the conformational suite of the HCAs in aqueous solution examined in the CD/anisotropy experiments in the present study can be understood as a first-order approximation of those found in liquid-like water-dominated ices. In addition, we compare experimentally recorded anisotropy spectra of HCAs with the ones of amino acids studied in similar environmental conditions[53,61], which suggest the potential role of stellar CPL for the *systematic* generation of enantiomeric excess across molecular families.

## Results and discussion
### CD and anisotropy spectra of aliphatic chain hydroxy monocarboxylic acids.
Figure 1 shows the ECD and anisotropy spectra of D- and L-enantiomers (R- and S-, respectively) of aliphatic chain HCAs, namely lactic, 2-hydroxybutanoic, 2-hydroxy-3-methylbutanoic and 2-hydroxy-4-methylpentanoic acids, in an aqueous solution in the 170–280 nm wavelength range. Confirmation of the mirror symmetry of CD spectra of respective enantiomers, i.e., nearly equal magnitude and position of extrema as seen in Fig. 1e is a valuable tool for demonstrating the reliability of recorded data. Despite the relatively high purity of the standards, differences in the absorbance spectra between the enantiomers of 2-hydroxybutanoic, 2-hydroxy-3-methylbutanoic

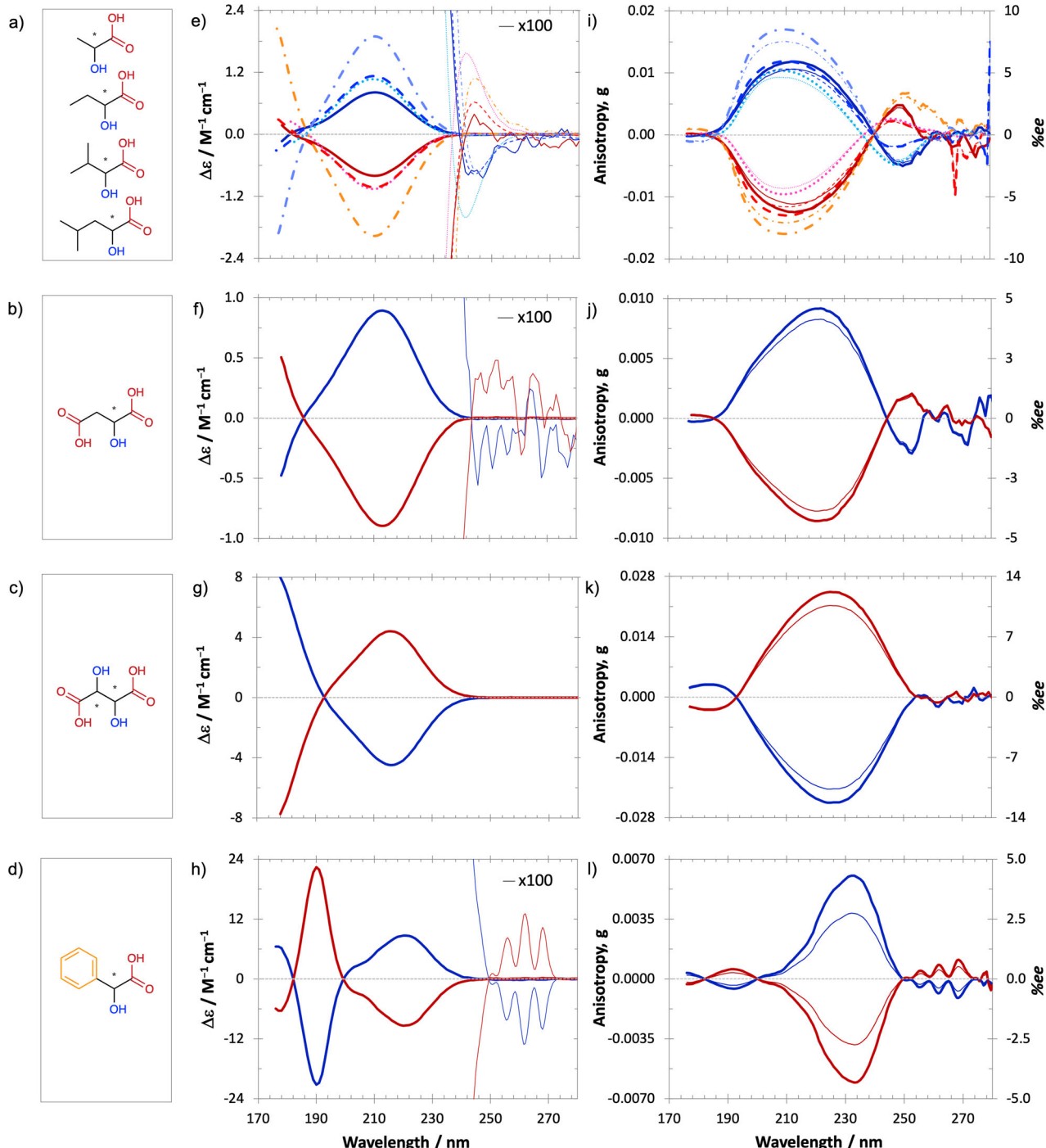

**Fig. 1 CD and anisotropy spectra of chiral hydroxycarboxylic acids in aqueous solution.** Schematic structures of **a** lactic, 2-hydroxybutanoic, 2-hydroxy-3-methylbutanoic and 2-hydroxy-4-methylpentanoic; **b** malic; **c** tartaric and **d** mandelic acid. **e–h** Corresponding electronic circular dichroism spectra of the HCAs; **e** lactic (solid dark blue/red line), 2-hydroxybutanoic (dashed blue/red line), 2-hydroxy-3-methylbutanoic (dotted turquoise/pink line), 2-hydroxy-4-methylpentanoic acid (dash-dotted purple/orange line). **i–l** Corresponding anisotropy spectra of the HCAs (thick) and the lower limit of the inducible enantiomeric excess (%ee) by either left- or right-circularly polarised light (thin) as a function of wavelength at the extent of reaction $\xi = 0.9999$ calculated based on the relation (1). The D-enantiomers (R-; except for tartaric acid where D-tartaric acid is 2S-, 3S-) are in shades of red and the L-enantiomers (S-; except for tartaric acid where L-tartaric acid is 2R-, 3R-) are in shades of blue. The collection of curves in (**e**) and (**i**) is shown separately for each compound in Supplementary Fig. S1.

and 2-hydroxy-4-methylpentanoic acids indicated the presence of absorbing contaminants. Since the ECD spectra of respective enantiomers were quasi-perfect mirror images, the presence of the impurities was reflected in the corresponding anisotropy spectra. Therefore, these were, for the L-2-hydroxybutanoic, D-2-hydroxy-3-methylbutanoic and D-2-hydroxy-4-methylpentanoic

acids corrected based on the absorbance of their optical antipodes of higher purity. Due to uncertainties induced by low absorbance and ECD signals, the values of the anisotropy factor g are less reliable above 260 nm for 2-hydroxybutanoic acid as well as above 270 nm for 2-hydroxy-3-methylbutanoic acid and D-2-hydroxy-4-methylpentanoic acid.

In the measured wavelength range, the ECD and anisotropy spectra of the L-enantiomers of all four aliphatic HCAs are dominated by a broad positive band with a maximum at around 210 nm (Fig. 1e and i, and Table 1). This ECD band was for lactic acid associated with the $n\pi^*$ transition of the carboxyl chromophore[45,51]. In addition to the broad band at ~210 nm, the CD spectra exhibit two negative features with minima below 180 nm and above 242 nm. While the former one for lactic acid was attributed to the $\pi\pi^*$ transition of the carboxyl group[45], the assignment of the latter one was more ambiguous. Toniolo et al. ascribed it solely to the $n\pi^*$ transition of the carboxyl chromophore[51], however, Craig and Pereira[46] reported that the associated transition involves coupling of one of the non-bonding orbitals of the oxygen atom attached to the chiral centre with the carbonyl chromophore. The apparent similarity of the ECD and anisotropy spectra of the four above-mentioned aliphatic chain HCAs stems from the fact that their structures differ only in the length/complexity of their aliphatic side chains (Fig. 1a). Since the side chains do not contain a strong chromophore and/or an additional chiral centre, they do not contribute significantly to the molecules' CD. Analogous behaviour was also observed for mono- and diamino carboxylic acids[62].

The deviations in the pH of the solutions and hence the equilibrium of analytes in different states of ionisation are likely to explain the minor shifts in the position and magnitude of CD maxima reported here and previously in the literature[43–45,51]. Note that the aqueous solutions of the HCAs investigated in the present study exhibit pH values between 2 and 3.1 (Table 1) at which most analytes are present in their undissociated protonated form.

The two negative CD bands of the L-enantiomers are a textbook example of how a relatively weak signal in the ECD spectrum can result in a relatively strong signal in the anisotropy spectrum, and vice versa. This highlights the importance of recording anisotropy spectra for assessing the effect of asymmetric photolysis by CPL. Given the significant drop in the photon flux density in the far UV range emitted by most stars (below about 177 nm)[31,63], the anisotropy spectra in Fig. 1i suggest that the sign and the magnitude of the resultant enantiomeric excess is most likely to be dictated by the anisotropy band at ~210 nm. Moreover, any stellar radiation with wavelengths below 200 nm is, in water-rich interstellar ices, likely to be absorbed by water molecules of the ice matrix to a high extent.

The UV-ECD and anisotropy spectra in an aqueous solution of amino acid counterparts of lactic, 2-hydroxy-3-methylbutanoic and 2-hydroxy-4-methylpentanoic acids—alanine, valine and leucine, respectively—are also dominated by a single broad band corresponding to the $n\pi^*$ transition of the carboxyl chromophore[35]. As it can be seen in Table 1 and Fig. 2, the dominant anisotropy bands in the aliphatic chain HCAs and their amino acid analogues are of the same sign. Yet, despite extensive investigations of interstellar ices via IR astronomical observations, we lack a complete understanding of their acidity and physical state of matter. However, recent CD and anisotropy spectroscopy experiments on essential amino acids in aqueous solution and ice matrices have revealed that the change in pH[61,64] and temperature[65] does not alter the sign of the CD/anisotropy bands, but only results in a slight shift (up to about 10 nm) and increase in the magnitude of the CD/anisotropy signals with decreasing pH and temperature. We therefore expect asymmetric photolysis driven by broadband stellar CPL of given helicity to induce enantiomeric excesses of the same handedness into aliphatic HCA and amino acid molecules in water-rich ices. Hence, the present data and the fact that lactic acid is known to be more resistant to racemisation[39] than its amino acid counterpart alanine could explain the reported chiral bias in lactic acid toward the L-enantiomer in meteoritic samples[39]. The fact that life uses

D-lactic and L-lactic acid is not in contradiction with the CPL scenario. Previous experiments have shown that the presence of L-enriched amino acids can bias the synthesis of sugars toward their D-forms under simulated prebiotic conditions[66]. This suggests that the initial chiral bias initiated by interstellar asymmetric photolysis does not necessarily resemble the one of present-day life due to possible bio- and geochemical racemisation and asymmetric catalytic processes on the early Earth.

Importantly, the present anisotropy spectra indicate that based on the CPL hypothesis not only lactic acid, but also the other aliphatic chain HCAs, are expected to occur in non-racemic ratios in meteoritic samples, especially in the most pristine ones. Most of the enantioselective analyses on lactic, 2-hydroxybutanoic and 2-hydroxy-3-mehtylbutanoic acids, so far, did not report on significant ee-s of these molecules in meteorites, which contrasts with their amino acid analogues alanine and valine with reported %$ee_L$ of up to 33.0% and 60.3% (Table 1), respectively. However, it should be noted that the number of reports on meteoritic amino acids is notably higher compared to HCAs, the significantly large %$ee_L$ values in amino acids are often critically discussed on the basis of potential terrestrial contamination, and recent studies[21,67] have found meteoritic amino acids in racemic ratios as well. In addition, the magnitudes of $ee_L$ induced in the aliphatic chain HCAs by stellar CPL are expected to be lower than %$ee_L$ uncertainties of enantioselective meteoritic analyses reported to date.

Although Peltzer and Bada[11] reported the presence of racemic lactic, 2-hydroxybutanoic and 2-hydroxy-3-methylbutanoic acids in the first enantioselective analyses of HCAs in the Murchison meteorite, the D/L values presented indicate a chiral bias toward the L-enantiomer. Notably, for lactic acid D/L = 0.92 ± 0.1 (corresponding to the %$ee_L$ range of −1 to 9.9%) and for 2-hydroxybutanoic acid D/L = 0.82 ± 0.1 (%$ee_L$ range of 4.2 to 16.3%). The results for 2-hydroxy-3-methylbutanoic acid are ambiguous due to the co-elution of an unknown compound with the L-enantiomer. Later on, Pizzarello et al.'s analyses[39] revealed racemic 2-hydroxybutanoic and 2-hydroxy-3-methylbutanoic acids in the Murchison, GRA 95229 and LAP 02342 meteorites. Unfortunately, the paper does not state exact %$ee_L$ values and/or uncertainties, only the approximate errors of around ±5% on concentrations of enantiomers are provided. The latest results of Aponte et al.[41] showed lactic and 2-hydroxybutanoic acids to be racemic within the following $ee_L$ intervals of −0.2 ± 7.9 and −0.8 ± 6.8% in the MIL 09011 meteorite, and 0.0 ± 4.8 and −0.8 ± 6.8% in the MIL090657 meteorite, respectively. Based on the anisotropy spectra maxima ($g(\lambda)_{max}$) in Table 1, the inducible %$ee_L$ of 2-hydroxybutanoic and 2-hydroxy-3-methylbutanoic acids by monochromatic CPL at the extent of reaction 0.9999 are ≥5.3% and ≥4.7% at pH 2 and 2.1, respectively, and are expected to be lower at higher pH[61]. Moreover, stellar CPL, which is polychromatic, with a dominant UV emission wavelength range of the majority of stars[31,63] matching the wavelength range studied here, would induce a net ee lower than the above-mentioned %$ee_L$ values. In addition, the photodecomposition of amino acids at neutral pH can lead to the stereoselective production of their corresponding HCAs via deamination (Supplementary Fig. S2a)[68]. Consequently, the same handedness of the CD/anisotropy spectra of amino acids and HCAs translates to the preferential photodecomposition of D-amino acids and an excess formation of the corresponding D-HCAs, which would diminish the L-excess produced by direct interaction of HCAs with CPL. Besides, non-stereoselective deamination of amino acids, thought to be the major photodecomposition pathway at low pH (Supplementary Fig. S2b) and leading to racemic hydroxy acid photolysis products[65], would also reduce the net L-excess of HCAs. Therefore, without amplification the $ee_L$ inducible by polychromatic CPL in the studied wavelength range to 2-hydroxybutanoic and 2-hydroxy-3-methylbutanoic acids upon astrophysical conditions is

**Table 1 Comparison of the hydroxycarboxylic acids' anisotropy factors _g_ (extremum wavelength) in aqueous solutions of given pH, corresponding %$ee_L$ values at the extent of reaction $\xi = 0.9999$ calculated based on relation (1) and %$ee_{L, m}$ values detected in different meteoritic samples with those previously reported for amino acid analogues.**

| Hydroxycarboxylic acid | | Amino acid | |
|---|---|---|---|
| **Lactic** | $g_{212} = 0.012$ (pH 2.4) <br> %$ee_L \geq 5.3$ <br><br> %$ee_{L, m} = 3–12^a$, $rac^{b,c}$ (-0.2 ± 7.9[c], 0 ± 4.8[c]), $NE^d$ | **Alanine** | $g_{226} = 0.025$ (pH 2)[e] / $g_{221} = 0.014$ (pH NR)[f] <br> %$ee_L \geq 10.9^g$ / $6.2^g$ <br><br> %$ee_{L, m}$ = -3–7, 33[h]; 3.4 ± 4.7, 15 ± 3[i]; $rac$ (6.5 ± 17.6, 1.3 ± 5)[c]; 29.7, $rac$ (0.7, -3, 0.2)[j] |
| **2-OH-butanoic** | $g_{209} = 0.012$ (pH 2.0) <br> %$ee_L \geq 5.3$ <br><br> %$ee_{L, m} = rac^{a,b,c}$ (-0.8 ± 6.8[c], -2.0 ± 6.9[c]), $NE^d$ | | |
| **2-OH-3-CH₃-butanoic** | $g_{208} = 0.010$ (pH 2.1) <br> %$ee_L \geq 4.7$ <br><br> %$ee_{L, m} = rac^{a,b}$, $NE^d$ | **Valine** | $g_{228} = 0.024$ (pH 2)[e] / $g_{218} = 0.010$ (pH NR)[f] <br> %$ee_L \geq 10.5^g$ / $4.5^g$ <br><br> %$ee_{L, m}$ = 0–3[h]; 9.7 ± 7.8, 3.0 ± 2.1, 67 ± 1[i]; 40 ± 7, -9.3 ± 7.2[c]; 60.3, 12.9, $rac$ (3.2, 0)[j] |
| **2-OH-4-CH₃-pentanoic** | $g_{210} = 0.017$ (pH 2.2) <br> %$ee_L \geq 7.5$ <br><br> %$ee_{L, m} = NR$ | **Leucine** | $g_{220} = 0.025$ (pH 2)[e] / $g_{220} = 0.015$ (pH NR)[f] <br> %$ee_L \geq 10.9^g$ / $6.7^g$ <br><br> %$ee_{L, m} = 72^i$ |
| **Malic** | $g_{222} = 0.0092$ (pH 2.3) <br> %$ee_L \geq 4.1$ <br><br> %$ee_{L, m} = NE^{a,k}$ | **Aspartic** | $g_{222} = 0.0096$ (pH 2)[e] <br> %$ee_L \geq 4.3^g$ <br><br> %$ee_{L, m}$ = 14 – 59[h]; 34 ± 2, 20 ± 2, 67 ± 1[i], $rac$ (-2.7 ± 14.5, -1.7 ± 21.3)[c] |
| **Tartaric** | $g_{225} = -0.024$ (pH 2.3) <br> %$ee_L \leq -10.7$ <br><br> %$ee_{L, m} = rac^k$ | | |
| **Mandelic** | $g_{232} = 0.006$ (pH 3.1) <br> %$ee_L \geq 2.7$ <br><br> %$ee_{L, m} = NR$ | **Phenylalanine**[l] | $g_{228} = 0.011$ (pH 2)[e] / $g_{227} = 0.005$ (pH NR)[f] <br> %$ee_L \geq 4.9^g$ / $2.5^g$ <br><br> %$ee_{L, m} = NR$ |

_rac_ racemic, _NR_ not reported, _NE_ not enantioseparated.
[a]Data reported by Pizzarello et al.[39], however, with unknown statistical analysis applied for %$ee_L$ values.
[b]Data reported by Peltzer and Bada[11].
[c]Data reported by Aponte et al.[41].
[d]Data reported by Pizzarello et al.[42].
[e]Data reported by Nishino et al.[61].
[f]Data reported by Bredehöft et al.[53].
[g]Calculated based on the _g_ values[53,61] using relation (1).
[h]Data reported in Glavin et al.[21].
[i]Data reported by Glavin et al.[76].
[j]Data reported by Furusho et al.[67]. Note that the data were reported in %L without errors. The %$ee_L$ values in the table were calculated based on the amounts of amino acid enantiomers reported in Table 2 in Furusho et al.[67].
[k]Data reported by Cooper et al.[71] with detection uncertainty not stated.
[l]Approximate analogy.

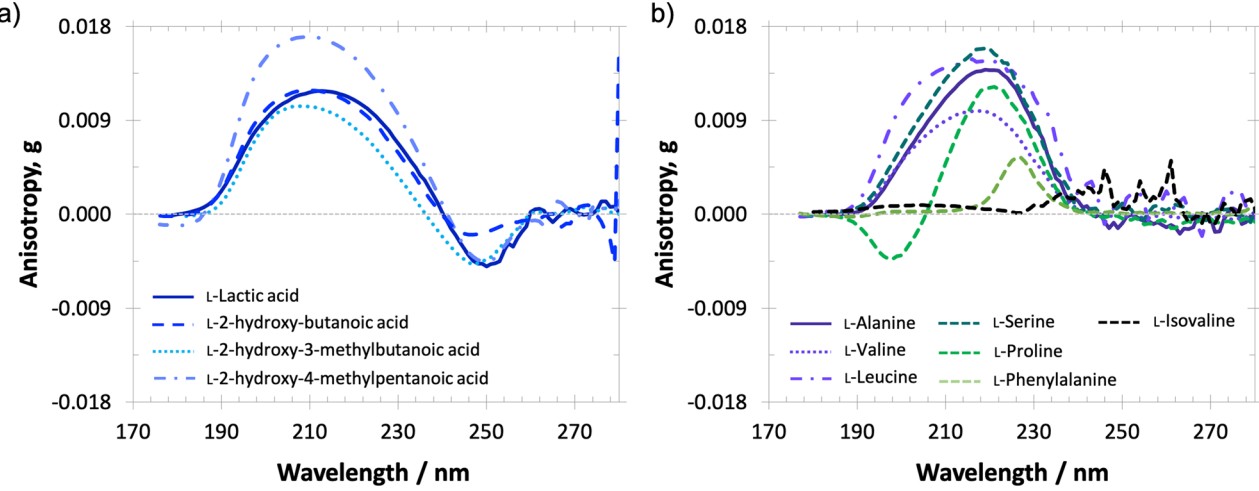

**Fig. 2 Comparison of the chiroptical properties of two distinct families of chiral molecules detected in meteorites.** Anisotropy spectra of **a** L-hydroxycarboxylic acids and **b** L-amino acids in aqueous solution. The same sign in the 170–280 nm wavelength range, which coincides with the dominant UV emission wavelength range of the majority of normal stars[31,63], indicates the same handedness of induced *ee* following the interaction with broadband CPL. Permission to reproduce the original amino acid data[53] shown in (**b**) is acknowledged, © 2014 Wiley Periodicals, Inc.

likely to be below the detection limits of the above-mentioned studies. The present anisotropy spectra therefore highlight the need for developing more sensitive analytical procedures for investigating enantiomeric composition of aliphatic chain HCAs in extra-terrestrial samples.

**CD and anisotropy spectra of hydroxy dicarboxylic acids**. The CD and anisotropy spectra of malic acid, (Fig. 1f and j, respectively), which is a C4 hydroxy dicarboxylic acid (Fig. 1b), resemble those of the aliphatic hydroxy monocarboxylic acids but with a slight redshift. The dominant CD band maximum is centred at 213 nm, while the anisotropy band peaks at 222 nm due to the shift in the positions of CD and absorption bands maxima[69]. The dominant CD and anisotropy spectral bands of tartaric acid (Fig. 1g and k, respectively) show opposite sign compared with malic acid, which demonstrates how a change in the chiral surrounding of a chromophore—replacing an H atom on the $\beta$-carbon by a hydroxy group—can exert symmetry-breaking perturbations of the electronic states and reverse the sign of the rotational strength, and consequently of the corresponding CD band. Compared with malic acid, the dominant CD and anisotropy bands of tartaric acid exhibit a redshift to 216 and 225 nm, respectively. The dominant CD bands were for both acids attributed to the $n\pi^*$ transition of the carboxyl chromophore[44,48,70].

Malic acid has been found in carbonaceous chondrites, however, its poor enantio-separation did not allow the determination of its enantiomeric composition[39,71]. To our knowledge, the only mention of tartaric acid detected in extra-terrestrial samples was in the study by Cooper et al.[71], who reported its presence in racemic ratio in the Murchison and Murray meteorites, however, without indicating the analyses' detection limits. While both acids naturally occur in their L-form in the biosphere, the opposite sign of the anisotropy spectra of tartaric acid compared with malic acid in the 170–280 nm wavelength range (Table 1) suggests that broadband CPL which would yield L-excess in amino acids and also in malic acid, would, in contrast, induce relatively high D-excess in tartaric acid under the studied environmental conditions. It is also worth noting that the peak anisotropy values for tartaric acid is the largest among the HCAs in the present study, further amplifying the expected observable excess (for comparison of maximum $ee_L$ values see Table 1). Since this molecule is present on Earth preferentially in its L-form, the

finding of D-excess in meteorites would remove ambiguities about potential terrestrial contamination. Hence future analyses of enantiomeric composition of tartaric acid in extra-terrestrial samples could provide deeper insights into what extent the medium- and pH-dependent equilibrium suite examined in the present study is of astrobiological relevance and help direct further studies on the effect of environmental conditions on the chiroptical response of prebiotic molecules.

**CD and anisotropy spectra of mandelic acid**. Mandelic acid's ECD spectrum (Fig. 1h) significantly differs from the above-mentioned HCAs due to the presence of a strong aromatic chromophore (phenyl) in its side chain (Fig. 1d), which contributes to the CD and anisotropy spectra in the near ultraviolet wavelength region (>250 nm, $\pi\pi^*$ excitation[49,52]) as well as in the vacuum ultraviolet (VUV) region. The VUV excitations can be classified as follows: 177 nm $n\pi^*$ excitation with charge transfer character from the carboxylic group, 190 nm $\pi\pi^*$ excitation with charge transfer character to the carboxylic group and 205 nm $\pi\pi^*$ excitation[49,52]. Although the band at 190 nm has the highest intensity in the ECD spectrum of mandelic acid, a weak oscillator strength in combination with a strong rotational strength of the $n\pi^*$ transition of the carboxyl chromophore[49] (CD band at 221 nm) causes this band to dominate in the anisotropy spectrum. Compared with aliphatic chain HCAs, the presence of the phenyl group instead of the aliphatic chain induces a bathochromic shift of the $n\pi^*$ transition in mandelic acid. Given the decline in UV photon flux of the majority of stars in the far UV range (<177 nm)[31,63] and strong UV absorption of water/ice matrix below 200 nm in water-rich interstellar ices[65], the active anisotropy band with the maximum at around 232 nm would dictate the enantiomeric excess resulting from the CPL irradiation. It should be noted that the anisotropy spectrum of phenylalanine[36] resembles the one of mandelic acid, hence the expected enantiomeric excess of the two molecules in examined environmental conditions induced by CPL would be similar and of the same handedness as in the aliphatic amino acids, however of smaller magnitude (for comparison of maximum $ee_L$ values see Table 1). Considering the relatively small anisotropy factors of mandelic acid and phenylalanine, and current analytical uncertainties, it is not surprising that their *ee*-s in meteoritic samples have not been reported yet.

## Conclusion

The potential role of UV CPL in generating enantiomeric enrichment of chiral precursors of life is an intriguing research question, which in order to be answered requires a synergistic approach of combining anisotropy spectroscopy and CPL irradiation experiments with the analyses of extra-terrestrial samples and astronomical observations. It is well documented that water-dominated interstellar ices play a pivotal role in the chemical and molecular evolutionary processes in molecular clouds and outer regions of protoplanetary disks. UV-irradiated interstellar ice analogues were previously found to exhibit liquid-like behaviour, which is likely to enhance the formation of complex organic species. This makes UV-irradiated water-rich interstellar ices strong candidates for the synthesis of essential molecules to kick-start the origin of life in a hospitable environment. Therefore, examining the chiroptical response of chiral organic species in aqueous solution to simulate a polarised solvation shell as closest model for water-rich interstellar ices represents an important initial step in elucidating the potential role of CPL in the evolution of biological homochirality. The present paper demonstrates that CPL irradiation of aliphatic chain HCAs in aqueous environments in the 170–280 nm wavelength range would yield the same handedness of enantiomeric excess as in amino acid counterparts. Our anisotropy spectroscopy experiments could thus explain the chiral bias toward the L-enantiomer of lactic acid previously detected in meteoritic samples. The experiments suggest that in the examined environmental conditions, 2-hydroxybutanoic, 2-hydroxy-3-methylbutanoic and 2-hydroxy-4-methylpentanoic acids would also yield an L-excess, however, the magnitudes are below the detection limits of previously reported enantioselective analyses of meteorites and hence it is not surprising that their excess has not been detected. On the contrary, the broadband CPL of the same helicity would yield D-excess in tartaric acid. Interestingly, tartaric acid is preferentially found in its L-form in the biosphere and hence it would be an excellent molecule to search for in extra-terrestrial samples, as it could provide deeper insights into the CPL scenario and more specifically into conditions in which CPL could have induced a prebiotic chiral bias. Analogously to monocarboxylic acids, alcohols and amines, the aromatic side chain HCA—mandelic acid—exhibits relatively small anisotropy factors which places even higher demands on the enantioselective analysis of extra-terrestrial samples. The present paper represents an important preparatory study for future analyses of extra-terrestrial samples from sample-return missions, notably the Hayabusa2 mission[72] which has recently successfully returned samples of 162173 Ryugu asteroid to Earth and the OSIRIS-REx[73] mission which has already collected and stowed samples of asteroid Bennu and is expected to return to Earth in September 2023. Moreover, optical rotation and circular dichroism spectroscopy are ideal tools as remotely or in situ accessible means of probing chirality as a biomarker for extra-terrestrial life[74]. Central within the application of chiroptical spectroscopy for future space missions will be to anticipate which chiral molecules and wavelength regions will have to be targeted and our anisotropy data hold potential information to develop such life detection strategies.

## Methods

Enantiopure standards of HCAs (lactic, 2-hydroxybutanoic, 2-hydroxy-3-methylbutanoic, 2-hydroxy-4-methylpentanoic, malic, tartaric and mandelic) were purchased from Sigma-Aldrich (purity ≥97%, D-lactic acid ≥90%) and were used without further purification. The aqueous solutions of HCAs were prepared in deionised, filtered, and UV-irradiated ultrapure water at concentrations between $0.4–125 \, g \, L^{-1}$.

UV ECD and anisotropy spectra of aqueous solutions of the HCAs were recorded in the 170–280 nm wavelength range using the SRCD facilities at the ASTRID2 synchrotron storage ring facility (Aarhus University, Denmark). A detailed description of the experimental set-up can be found in refs. [36,53]. Differential absorption and absorption spectra were measured simultaneously as described in[69]. Quartz cells with a nominal pathlength of 100 $\mu$m were used for the experiments, with the pathlength of each cell carefully measured using an interference technique[75]. The chiral response of HCAs in aqueous solution was examined at different concentrations to ensure good signal-to-noise ratio in the whole measured wavelength range. The spectra were recorded with a 2 s dwell time and multiple accumulations for each enantiomer. For most spectra the wavelength steps were 1 nm, but smaller steps were used to capture the long wavelength fine structures of mandelic acid. A mild 7-point window Savitzky-Golay filter smoothing was applied to the CD spectra and therefore also the resulting anisotropy spectra.

## Data availability

All data analysed during this study are included in this published article and its supplementary information files. The datasets generated or analysed during the current study are available from the corresponding authors upon request.

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

## Acknowledgements

This research has received funding from the European Research Council under the European Union's Horizon 2020 research and innovation programme [grant agreement 804144]. Further funding was provided by the French government through the UCA$^{JEDI}$ Investments in the Future project managed by the National Research Agency (ANR) with the reference number ANR-15-IDEX-01, by the ANR under grant number [ANR-18-CE29-0004-01] as well by the project CALIPSOplus under the Grant Agreement 730872 from the EU Framework Programme for Research and Innovation HORIZON 2020.

## Author contributions

This study has been carried out through equal contributions of all authors. J.B. and C.M. wrote the paper and N.C.J., U.J.M., and S.V.H. have given approval to the final version of the manuscript.

## Competing interests

The authors declare no competing interests.
