## [Peer Review File · Communications Chemistry]

Reviewers' comments:

Reviewer #1 (Remarks to the Author):

Bocková et al. extends the groups anisotropy spectroscopy to hydroxy acids. This is an important line of research, as circularly polarized photolysis or synthesis is a popular, yet still insufficiently supported mechanism to break molecular symmetry in meteoritic organics and presumably on the ancient Earth (or possibly Mars). Only amino acids and aldonic acids and one deoxyaldonic acid (or hydroxy acid, lactic acid) have been detected to break symmetry in meteorites. This paper focuses on predictions for asymmetry in hydroxy acids based on anisotropy spectroscopy. This work is likely to be an important resource for modelers and experimentalists studying one of the most important questions related to the origin of life: the origin of homochirality.

- As stated the Pizzarello et al. 2010 values are given as 3-12% with an approximate 5% error. This is more significant error than it sounds at first. This error is on the individual measurements (which works out to the Lee error as well), not a 5% error on the Lee value. Thus, the 3% Lee is $3\% \pm 5$ (i.e. -2% to 8%), not $3 \pm 5\%$ (i.e. not 2.85% to 3.15%). Specifically:

$\%Lee = (L-D)/(L+D)*100$, so for GRA 95299 where the reported values in nmol/g are $D=16.8, L=17.8$
The +5% error case is $(16.91-17.64)/(16.91+17.64)*100 = -2.1$ and $(15.96-18.69)/(15.96+18.69)*100 = +7.9$

Thus, GRA 95229 has an Lee between +7.9 and -2.1 (racemic within error), LAP 02342 Lee between +1.1 to +11, and Murchison +0.23 to +10 (pretty much racemic within error).

It is not clear how to reconcile the parenthetical ranges for Murchison in Table 1 from the analysis of other Murchison stones. That is, were the values for Murchison $D=28.3, L=32.0$ ($\%Lee=6.1 \pm 5$), $D=58.8, L=64.5$ ($\%Lee=4.6 \pm 5$, i.e. racemic) and the average of three measurements the $D=39.3, L=44.3$ ($\%Lee=5.2 \pm 5$)? Since no value was given in Table 1 that produces an Lee of 12%. One hopes that the 12% maximum Lee is the upper limit of the error for LAP 02342 of 11.06% accidentally "rounded up" to 12%. Without asking the authors, there is no way to know.

Regardless, of the three meteorites reported it is not clear which exhibited 12% Lee and the greatest Lee explicitly given was in LAP 02342 at $\%Lee=6.1 \pm 5$

Also, please note that 70-80% lost during drying. This level of loss could have resulted in isotopic fractionation which could have masked contamination. The samples did show alanine contamination, so lactic contamination is not outrageous. Also, Pizzarello et al. 2010 express concern that the $\delta^{13}C$ values for Murchison in Table 3 for lactic acid are suspect, due to the above evaporation losses during workup, but never discuss the more likely fractionation to enrich δD values.

This is a long way of saying to be cautious in relying too heavily on this report of enantiomeric excesses of lactic acid. So the level of exuberance over lactic acid asymmetry in the manuscript, including the conclusions needs to be tempered. I strongly recommend taking the opportunity in this manuscript to more honestly present the lactic acid results, at a minimum by explicitly giving the error as $\%Lee$ of 3 ± 5 to 12 ± 5 , which is the most charitable interpretation.

- Also, note that racemic lactic acid was detected in MIL 090001 (-0.2±7.9) and MIL 090657 (0.0±4.8) in Aponte et al. 2020 (doi: 10.1111/maps.13586). This reference, and that errors are calculated and given, should be included in the manuscript. That lactic acid has modest meteoritic Lee or is racemic within relatively large errors is not a detriment to the manuscript, but should challenge meteoriticists to improve their methods to drive the detection errors of lactic acid down.
- Fig 1e and i are very hard to understand and the caption does not adequately label which curves are which without looking in the supplemental. Please add Fig S1 to the main document instead of in the supplemental and remove boxes a, e, and I from Fig 1.
- Why was mandelic acid studied, when the hydroxy acid equivalent of phenylalanine is 3-phenyllactic acid, which is commercially available and inexpensive? If it was selected, not as an amino acid analogue, but as a chromophore then the phenylalanine in Table 1 needs to be explained as an approximate analogy.
- It is unfortunate that Table 1 is lacking α -amino-n-butyric acid and diamino succinic acid, which though not critical for the manuscript would improve the symmetry of the figure. Please also include the lactic acid and alanine structures in Table 1.

Reviewer #2 (Remarks to the Author):

Meinert and co-workers report the ECD spectra of a series of chiral alpha-hydroxy carboxylic acids and discuss the potential that these molecules could be formed in space (leading to discussions on origin of homochirality and showing perspectives for data analysis in context of meteorite research). The authors show that the anisotropy spectra of the hydroxy carboxylic acids are very similar to those of amino acids, so that parallels in their photochemical behaviour can be expected. The data and analysis is well presented and thus certainly deserve publication. Nevertheless, I am not convinced of the particular importance respectively impact of the study. I must say, I am not informed about the challenges associated with synchrotron CD measurements, but as the authors did not discuss this in any way I assume it is straightforward once one has access to a synchrotron facility. Then, however, the paper simply reports a few CD spectra embedded in extensive literature discussion and band assignments. On p. 4, l. 124, the authors list a few papers with band assignments, some of them "ambiguous" (l. 130) - some theoretical support would have been nice here. Generally, after the authors have highlighted parallels between hydroxy and amino acids, a comparison based on spectra calculations would have been nice as well. I understand that one wants to verify band intensities experimentally, but it would be good to know how far off electronic structure calculations are in predicting the intensities. In conclusion, a nice study which I would probably recommend for publication with major revisions (due to missing calculations) to a more specialized journal, but not for Nature Comms Chem.

Reviewer #3 (Remarks to the Author):

This paper reports the results of laboratory experiments on the effect of circularly polarised light (CPL) on an inducible L- and D-enantiomeric excess (ee) of hydroxycarboxylic acids (HCAs). The conclusion is that the extent of the inducible enantiomeric excess found in these experiments was

small and thus consistent with meteorite analyses reported previously. Both the main theme and the conclusion of this study are important in prebiotic chemistry and astrobiology. However, the discussion on this experiment and the interstellar CPL is not sufficient, and this discussion may affect the conclusion. Therefore, I recommend the editor publish this manuscript in Communications Chemistry after the major revision.

Major comments:

The conclusion is supported by the result of the maximum ee found in this experiment. All simulation experiments have differences from natural events. In the present case, strength, duration, wavelength, and others should differ from the natural CPL. The author should discuss the effects of these differences on the ee found in this study.

Recently, a new report found the absence of amino acid ee in meteorites. The author should include this in the discussion or introduction.

A. Furusho, T. Akita, M. Mita, H. Naraoka, K. Hamase, Three-dimensional high-performance liquid chromatographic analysis of chiral amino acids in carbonaceous chondrites., *Journal of Chromatography A*, doi.org/10.1016/j.chroma.2020.461255, 1625, 461255 (8pp), 2020.05.

The author should discuss whether this explains the difference of ee between amino acids and HCAs found in meteorites. Why amino acids show large ee and HCAs show small?

We provide here point-by-point responses to the reviewers' reports. The reviewers' reports are copied below in *italics* and our responses are inserted in plain text. Changes in the manuscript have been highlighted in grey.

Reviewer #1

Bocková et al. extends the groups anisotropy spectroscopy to hydroxy acids. This is an important line of research, as circularly polarized photolysis or synthesis is a popular, yet still insufficiently supported mechanism to break molecular symmetry in meteoritic organics and presumably on the ancient Earth (or possibly Mars). Only amino acids and aldonic acids and one deoxyaldonic acid (or hydroxy acid, lactic acid) have been detected to break symmetry in meteorites. This paper focuses on predictions for asymmetry in hydroxy acids based on anisotropy spectroscopy. This work is likely to be an important resource for modelers and experimentalists studying one of the most important questions related to the origin of life: the origin of homochirality.

Comment #1-1: *As stated the Pizzarello et al. 2010 values are given as 3-12% with an approximate 5% error. This is more significant error than it sounds at first. This error is on the individual measurements (which works out to the Lee error as well), not a 5% error on the Lee value. Thus, the 3% Lee is 3%±5 (i.e. -2% to 8%), not 3±5% (i.e. not 2.85% to 3.15%). Specifically:*

*%Lee = (L-D)/(L+D)*100, so for GRA 95299 where the reported values in nmol/g are D=16.8,L=17.8 The +5% error case is (16.91-17.64)/(16.91+17.64)*100 = -2.1 and (15.96-18.69)/(15.96+18.69)*100= +7.9*

Thus, GRA 95229 has an Lee between +7.9 and -2.1 (racemic within error), LAP 02342 Lee between +1.1 to +11, and Murchison +0.23 to +10 (pretty much racemic within error).

It is not clear how to reconcile the parenthetical ranges for Murchison in Table 1 from the analysis of other Murchison stones. That is, were the values for Murchison D=28.3,L=32.0 (%Lee=6.1±5), D=58.8,L=64.5 (%Lee=4.6±5, i.e. racemic) and the average of three measurements the D=39.3,L=44.3 (%Lee=5.2±5)? Since no value was given in Table 1 that produces an Lee of 12%. One hopes that the 12% maximum Lee is the upper limit of the error for LAP 02342 of 11.06% accidentally "rounded up" to 12%. Without asking the authors, there is no way to know.

Regardless, of the three meteorites reported it is not clear which exhibited 12% Lee and the greatest Lee explicitly given was in LAP 02342 at %Lee=6.1±5

Also, please note that 70-80% lost during drying. This level of loss could have resulted in isotopic fractionation which could have masked contamination. The samples did show alanine contamination, so lactic contamination is not outrageous. Also, Pizzarello et al. 2010 express concern that the δ13C values for Murchison in Table 3 for lactic acid are suspect, due to the above evaporation losses during workup, but never discuss the more likely fractionation to enrich δD values.

This is a long way of saying to be cautious in relying too heavily on this report of enantiomeric excesses of lactic acid. So the level of exuberance over lactic acid asymmetry in the manuscript, including the conclusions needs to be tempered. I strongly recommend taking the opportunity in this manuscript to more honestly present the lactic acid results, at a minimum by explicitly giving the error as %Lee of 3±5 to 12±5, which is the most charitable interpretation.

Also, note that racemic lactic acid was detected in MIL 090001 (-0.2±7.9) and MIL 090657 (0.0±4.8) in Aponte et al. 2020 (doi: 10.1111/maps.13586). This reference, and that errors are calculated and given, should be included in the manuscript. That lactic acid has modest meteoritic Lee or is racemic within relatively large errors is not a detriment to the manuscript but should challenge meteoriticists to improve their methods to drive the detection errors of lactic acid down.

Response #1-1: Thank you very much for this constructive comment and your thorough elaboration. We agree that Pizzarello's report does not clearly explain how the %ee_L range of 3 – 12% on lactic acid was calculated and we also fully agree that we should emphasise this to the readers of our manuscript more clearly since this may cast doubts on potential contamination, and hence the reliability of the data.

The main reason why we did not calculate and state explicitly the errors of the %ee_L values reported in Pizzarello's study, but rather reported the quantification uncertainties of the concentrations of ±5% is that without knowing the approach used by Pizzarello et al. to calculate the range 3 – 12%, we cannot do it correctly. Notably:

- Based on Pizzarello's study the 3 and 12% indicate the minimum and maximum values of detected %ee_L values: "Lactic acid was found to display L-ee of various amplitude (3–12%) in all samples (Fig. 2)" rather than the average values for which they did not state the errors. The ±5% error stated below Table 1 in Pizzarello's study is the error of the concentrations they determined (very likely just an approximation since it is hard to believe it would be exactly 5% for all of the measurements), and these would translate into the %ee_L error. However, not as ±5%, but the average values of concentrations should be used to calculate the average %ee_L and error propagation should be applied. We know that the 3 – 12% are not the average values of %ee_L as we can determine these based on the concentrations provided as 2.9% for the GRA 95229; 6.1% for the LAP 02342 and 5.2% (or 6.1%, or 4.6%) for the Murchison meteorites. Therefore, we consider it to be misleading to state 3±5% to 12±5%.
- The approach of calculating the %ee_L range by taking into account the minimum and maximum values of the concentrations of both enantiomers (e.g. average concentration ±5%) would represent the worst case scenario and would not enable the most accurate ee_L values to be retrieved from Pizzarello et al.'s study. This can be very well seen e.g. in the study of Aponte et al. 2020 (doi: 10.1111/maps.13586), where the concentrations are given with uncertainties as well as the calculated %ee_L. Notably, for MIL 09011 where the reported concentrations for lactic acid are D = 46.3±5.4 nmol/g and L = 46.5±4.9 nmol/g, the minimum %ee_L = (41.6 - 51.7)/(41.6 + 51.7)×100% = -10.8% and the maximum %ee_L = (51.4 - 40.9)/(51.4 + 40.9)×100% = 11.4%, hence the %ee_L range calculated based on the reported concentrations would be -10.8% to 11.4%, while the reported %ee_L by Aponte et al. is -0.2±7.9%, i.e. the range -8.1 to 7.7%. Therefore, by calculating the %ee_L by using the minimum/maximum concentrations, we would get higher uncertainties on %ee_L than the ones reported by Aponte *et al.* In the case of Pizzarello et al, this approach would lead to the %ee_L range of -2.1 to 11.1%, This does not necessarily mean that the 3-12% range reported by Pizzarello et al. is wrong, but that they did not use the approach of calculating the worst-case scenario we mention above, which is understandable since they had the full data set (similar to Aponte et al.). Therefore, we do not consider it to be appropriate to do our own calculation of the worst-case scenario and claim the 3 – 12% reported by Pizzarello et al. to be incorrect.
- Moreover, Pizzarello et al. did not necessarily calculate the %ee_L using concentrations but could have based their %ee_L calculations on the peak areas of both enantiomers in single/total ion chromatograms. Providing that for both enantiomers the concentration c scales with the peak area A as c = kx A, where k is a constant, they could have worked in ion counts instead. This way they would have minimised additional sources of uncertainty resulting from their calibration curves, i.e. translation from ion counts to nmol/g. In addition, calibration curves of standards and meteoritic samples are very likely to exhibit different slopes due to matrix effects, which if not considered can lead to incorrect concentration values.

- Alternatively, they could have worked up the absolute concentrations and calculate %ee_L for each separate analysis.

In summary, it is very unfortunate that Pizzarello et al. did not comment on their data treatment and statistical analysis applied for calculating the range of %ee_L values and we can only hope that the peer-reviewers of the paper confirmed with them the statistical analysis behind the reported 3–12%, since this is one of the key outcomes of the paper. Notwithstanding, to not report false %ee_L ranges for lactic acid from Pizzarello’s study in our manuscript, we revised our manuscript in the way that we clearly state all of the above-mentioned limitations of Pizzarello’s study including unknown data treatment and sample loss of up to 80%. We have also tempered the statements claiming that “our results on lactic acid are in agreement with the results of Pizzarello’s study which show L-excess of 3–12 %” throughout the manuscript, so that the manuscript does not give the reader a false impression of high reliability of Pizzarello et al.’s data.

Specific changes related to this comment can be found here:

- line 17,
- lines 70-75,
- removed sentence “Nevertheless, only lactic acid was found in a significant L-ee in Murchison.....”,
- line 72 / lines 216-218 / Table 1 / line 301: added ref. Aponte *et al.* (2020), doi: 10.1111/maps.13586,
- line 72 / Table 1 / line 302: added ref. Pizzarello *et al.* (2012), doi.org/10.1073/pnas.1204865109, in which lactic acid, 2-OH-butyrinic and 2-OH-3-CH₃-butyrinic acids were not enantioseparated,
- line 180,
- line 206-215,
- line 299,
- line 334-335.

Importantly, we agree that the uncertainties over %ee_L values in the reported meteoritic studies do not in any way diminish the relevance/importance of the outcomes of the present manuscript, but we rather believe that the manuscript will be understood as a guide for improving the current enantioselective analytical procedures applied to study extra-terrestrial samples.

Comment #1-2: *Fig 1e and i are very hard to understand and the caption does not adequately label which curves are which without looking in the supplemental. Please add Fig S1 to the main document instead of in the supplemental and remove boxes a, e, and I from Fig 1.*

Response #1-2: Thank you for pointing this out. We added the descriptions of which line corresponds to which HCA to the corresponding caption, notably (lines157-159):

“lactic (solid dark blue/red line), 2-hydroxybutanoic (dashed blue/red line), 2-hydroxy-3-methylbutanoic (dotted turquoise/pink line), 2-hydroxy-4-methylpentanoic acid (dash-dotted purple/orange line)”

However, we believe that keeping the circular dichroism (CD) and anisotropy spectra of lactic, 2-hydroxybutanoic, 2-hydroxy-3-methylbutanoic and 2-hydroxy-4-methylpentanoic acids in the same slots of Fig. 1e and i, respectively, enables the reader to readily compare these with the ones of malic, tartaric and mandelic acids in the same Figure. We opted for joint slots for the CD/anisotropy spectra of structurally related lactic, 2-hydroxybutanoic, 2-hydroxy-3-methylbutanoic and 2-hydroxy-4-methylbutanoic acids because they are extremely similar. In addition, the wavelengths corresponding to the maxima in the anisotropy spectra are stated in Tab. 1. Therefore, we hope that the corrected labelling would be sufficient to guide the reader in Fig. 1e and i and more details are

provided in the supplementary materials, where we present the full spectra as they appear in Fig. 1e and i one by one.

Comment #1-3: *Why was mandelic acid studied, when the hydroxy acid equivalent of phenylalanine is 3-phenyllactic acid, which is commercially available and inexpensive? If it was selected, not as an amino acid analogue, but as a chromophore then the phenylalanine in Table 1 needs to be explained as an approximate analogy.*

Response#1-3: Absolutely, the selection of mandelic acid was based on the chromophore and we added a note below Tab. 1 indicating that phenylalanine is only an approximate analogy (line 311).

Comment #1-4: *It is unfortunate that Table 1 is lacking α -amino-n-butyric acid and diamino succinic acid, which though not critical for the manuscript would improve the symmetry of the figure.*

Response #1-4: We applied the following major criteria for the selection of hydroxy acids studied in the present paper:

- Analogues of the amino acids for which the CD/anisotropy spectra have already been reported.
- Systematic changes in the side chains.
- The molecules have been detected in meteorites.

2-OH-butanoic and tartaric acids were selected based on the points 2 and 3. Even if we do not know the CD/anisotropy spectra of their amino acid analogues α -amino-n-butyric acid (2-Aba) and diamino succinic acid, they represent an important contribution to the paper.

Comment #1-5: *Please also include the lactic acid and alanine structures in Table 1.*

Response #1-5: Thank you for highlighting this editing mistake. We added both structures to Table 1.

Reviewer #2

Meinert and co-workers report the ECD spectra of a series of chiral alpha-hydroxy carboxylic acids and discuss the potential that these molecules could be formed in space (leading to discussions on origin of homochirality and showing perspectives for data analysis in context of meteorite research). The authors show that the anisotropy spectra of the hydroxy carboxylic acids are very similar to those of amino acids, so that parallels in their photochemical behaviour can be expected. The data and analysis is well presented and thus certainly deserve publication. Nevertheless, I am not convinced of the particular importance respectively impact of the study. I must say, I am not informed about the challenges associated with synchrotron CD measurements, but as the authors did not discuss this in any way I assume it is straightforward once one has access to a synchrotron facility. Then, however, the paper simply reports a few CD spectra embedded in extensive literature discussion and band assignments. On p. 4, l. 124, the authors list a few papers with band assignments, some of them "ambiguous" (l. 130) - some theoretical support would have been nice here. Generally, after the authors have highlighted parallels between hydroxy and amino acids, a comparison based on spectra calculations would have been nice as well. I understand that one wants to verify band intensities experimentally, but it would be good to know how far off electronic structure calculations are in predicting the intensities. In conclusion, a nice study which I would probably recommend for publication with major revisions (due to missing calculations) to a more specialized journal, but not for Nature Comms Chem.

Response #2: Thank you for your opinion on the manuscript. You have raised an important point about the importance or impact of the study. In the following, we underline a few key outcomes of our study and highlight the impact of these findings in the field of the origin-of-life. Notably, hydroxycarboxylic acids are broadly considered to have been key molecules at the dawn of life given their potential role in building proto-peptides along with amino acids as well as primitive compartments preceding membranes. Moreover, homochirality of life is a key biosignature and the circularly polarised light (CPL) scenario is one of the most promising deterministic scenarios for explaining the symmetry breaking event, so far. Therefore, to unravel the potential role of CPL, the first step is to record the corresponding anisotropy spectra. The present manuscript is the first one to report the anisotropy spectra of structurally diverse hydroxycarboxylic acids. Importantly, the manuscript does not simply report on the anisotropy spectra and the maximum inducible enantiomeric excesses in asymmetric photolysis experiments, but for the first time discusses the effect of the more relevant broadband interstellar CPL on hydroxy acids in comparison with amino acids. This is an important novelty since while it has been shown that CPL can induce a chiral bias in amino acids and several other molecules, the relevance of the CPL scenario can only be understood when investigating the response of all homochiral biomolecules and comparing these results with their potential excess in meteoritic samples. This paper, therefore, highlights the alike response of hydroxy acids and amino acids to CPL which agrees with the bias toward L-enantiomers in amino acids and lactic acid in meteoritic samples. In addition, the manuscript highlights the relatively low magnitudes of CPL inducible enantiomeric excesses which are not within the detection capabilities of currently used enantioselective analytical procedures applied to study meteoritic samples. Therefore, the present manuscript can be considered as an important guide for astrobiologists and scientists analysing meteorites to improve their detection uncertainties as well for preparing the analytical procedures for the on-going and future sample return missions, notably Hayabusa 2 and OSIRIS-REx. This is crucial for reaching a common goal of elucidating the chiral force responsible for symmetry breaking and the origin of life on Earth.

Thank you for expressing your interest in seeing the results of theoretical calculations simulating our experimental circular dichroism spectra of hydroxycarboxylic acids. We agree that the strong experimental data set on the selection of HCAs with various side chains examined in our study is an ideal test case for assessing how good the current calculations are and even more for potentially improving the applied theoretical approaches. Notwithstanding, in the case of our study this would be out of the scope since such calculations would be a study on its own. A recent paper by Evans et al. 2021 (doi.org/10.1039/D0RA06832B) compares their electronic circular dichroism spectra of

asymmetric small molecules with theoretical calculations (time-dependent density functional theory in combination with polarisable continuum model for simulating solvent effects and a thermal averaging over molecular conformations) and highlights the limitations of their calculations in accurately predicting the ECD spectra. Hence, to carry out and report a thorough theoretical study for accurately predicting the ECD spectra of HCAs in our manuscript would exceed the scope of our manuscript. Moreover, such calculations would not improve the reliability of the presented data or improve the interpretation of the presented data or change the findings of the present manuscript. On the contrary, the experimental data set would rather serve as a standard for improving and assessing the quality of calculations. The consistency of the experimental CD spectra is confirmed by the mirroring effect observed for each pair of enantiomers. Therefore, we are delighted that you raised this point as this suggests that our manuscript would find interest not only within the origin-of-life community but can be used as a reference for theoreticians and is likely to inspire further theoretical studies.

Reviewer #3

This paper reports the results of laboratory experiments on the effect of circularly polarised light (CPL) on an inducible L- and D-enantiomeric excess (ee) of hydroxycarboxylic acids (HCAs). The conclusion is that the extent of the inducible enantiomeric excess found in these experiments was small and thus consistent with meteorite analyses reported previously. Both the main theme and the conclusion of this study are important in prebiotic chemistry and astrobiology. However, the discussion on this experiment and the interstellar CPL is not sufficient, and this discussion may affect the conclusion. Therefore, I recommend the editor publish this manuscript in Communications Chemistry after the major revision.

Comment #3-1: *Major comments: The conclusion is supported by the result of the maximum ee found in this experiment. All simulation experiments have differences from natural events. In the present case, strength, duration, wavelength, and others should differ from the natural CPL. The author should discuss the effects of these differences on the ee found in this study.*

Response #3-1: Thank you for your insightful comment. We agree that asymmetric photolysis experiments have their limitations in simulating the conditions in space, some of which you pointed out, however, these cannot be easily overcome mainly due to the lack of knowledge on the exact conditions in space and/or the feasibility of the experiments, e.g. in terms of time scales. As the present manuscript does not report on asymmetric photolysis experiments, we believe to have addressed some of the differences between the experiments in the laboratory and real conditions in space in the original manuscript, notably in the following sentences:

- Lines 148-153: “Given the significant drop in the photon flux density in the far UV range emitted by most stars (below about 177 nm)^{31,63}, the anisotropy spectra in Fig. 1i suggest that the sign and the magnitude of the resultant enantiomeric excess is most likely to be dictated by the anisotropy band at ~210 nm. Moreover, any stellar radiation with wavelengths below 200 nm is, in water-rich interstellar ices, likely to be absorbed by water molecules of the ice matrix to a high extent⁶⁴.”
- Lines 170-178: “Yet, despite extensive investigations of interstellar ices *via* IR astronomical observations, we lack a complete understanding of their acidity and physical state of matter. However, recent CD and anisotropy spectroscopy experiments on essential amino acids in aqueous solution and ice matrices have revealed that the change in pH^{61,65} and temperature⁶⁴ does not alter the sign of the CD/anisotropy bands, but only results in a slight shift (up to about 10 nm) and increase in the magnitude of the CD/anisotropy signals with decreasing pH and temperature. We therefore expect asymmetric photolysis driven by stellar broadband CPL of given helicity to induce enantiomeric excesses of the same handedness into aliphatic HCA and amino acid molecules in water-rich ices.”

However, we gratefully take this opportunity to add more details, notably:

- Lines 86-89: “Photons with energies of 5 to 9 eV (~ 248–138 nm) have mean free paths comparable to the thickness of interstellar ices (~0.01 μm thick), which suggests that the majority of the ice layers covering interstellar dust particles would evolve through photochemical reactions⁵⁴.”
- Lines 218–224: “Based on the anisotropy spectra maxima ($g(\lambda)_{\text{max}}$) in Tab. 1, the inducible % ee_L of 2-hydroxybutanoic and 2-hydroxy-3-methylbutanoic acids by monochromatic CPL at the extent of reaction 0.9999 are $\geq 5.3\%$ and $\geq 4.7\%$ at pH 2 and 2.1, respectively, and are expected to be lower at higher pH⁶¹. Moreover, stellar CPL, which is polychromatic, with a dominant UV emission wavelength range of the majority of stars^{31,63} matching the wavelength range studied here, would induce a net ee lower than the above-mentioned % ee_L values.”

To account for monochromatic CPL outside the wavelengths corresponding to the maxima of the anisotropy spectra or for lower extents of reaction and/or for polychromatic CPL in the given wavelength range, we removed “just at or” in the following sentence and also tempered the statement by replacing “would” for “is likely to”.

- Line 232-235: “Therefore, without amplification the ee_L inducible by polychromatic CPL in the studied wavelength range to 2-hydroxybutanoic and 2-hydroxy-3-methylbutanoic acids upon astrophysical conditions is likely to be below the detection limits of the above-mentioned studies.”
- Importantly, we have mentioned in the Conclusion of the original manuscript that the present study should be considered as an important starting point in elucidating the role of CPL in generating the chiral bias in space rather than considering this hypothesis as being proven:
- Lines 327-330: “Therefore, examining the chiroptical response of chiral organic species in aqueous solution to simulate a polarised solvation shell as closest model for water-rich interstellar ices represents an important initial step in elucidating the potential role of CPL in the evolution of biological homochirality.”
- In the Conclusion of the original manuscript, we claimed that in the examined conditions the magnitudes of the enantiomeric excess inducible in hydroxy acids based on the anisotropy data are close to the detection limits. There was clearly missing the word “maximum” (i.e. maximum magnitudes). However, to make it more general and account for monochromatic CPL outside the wavelengths corresponding to the maxima of the anisotropy spectra and/or for lower extents of reaction and/or polychromatic CPL in the given wavelength range we have now added the word “below” instead of “maximum”.
- Line 335-339: The experiments suggest that in the examined environmental conditions, 2-hydroxybutanoic, 2-hydroxy-3-methylbutanoic and 2-hydroxy-4-methylpentanoic acids would also yield an L-excess, however, the magnitudes are below the detection limits of previously reported enantioselective analyses of meteorites and hence it is not surprising that their excess has not been detected.

Comment #3-2: *Recently, a new report found the absence of amino acid ee in meteorites. The author should include this in the discussion or introduction. A. Furusho, T. Akita, M. Mita, H. Naraoka, K.*

Hamase, Three-dimensional high-performance liquid chromatographic analysis of chiral amino acids in carbonaceous chondrites., Journal of Chromatography A, doi.org/10.1016/j.chroma.2020.461255, 1625, 461255 (8pp), 2020.05.

The author should discuss whether this explains the difference of ee between amino acids and HCAs found in meteorites. Why amino acids show large ee and HCAs show small?

Response #3-2: Thank you for highlighting the reference “Furusho *et al.* 2020 (doi.org/10.1016/j.chroma.2020.461255)” to us. We have added the corresponding reference to the manuscript, notably in: line 202, 308-309 and Table 1.

In the original manuscript we have already included the following statement explaining that the enantioselective photodecomposition can diminish the enantiomeric excess induced in hydroxy acids, notably:

- Lines 224-232: In addition, the photodecomposition of amino acids at neutral pH can lead to the stereoselective production of their corresponding hydroxycarboxylic acids *via* deamination (Fig. S2a)⁶⁸. Consequently, the same handedness of the CD/anisotropy spectra of amino acids and HCAs translates to the preferential photodecomposition of D-amino acids and an excess formation of the corresponding D-HCAs, which would diminish the L-excess produced by direct interaction of HCAs with CPL. Besides, non-stereoselective deamination of amino acids, thought to be the major photodecomposition pathway at low pH (Fig. S2b) and leading to racemic hydroxy acid photolysis products⁶⁴, would also reduce the net L-excess of HCAs.

Referring to your concern on the difference between the enantiomeric excesses found in amino acids and hydroxy carboxylic acids, we have further added the following sentences:

- Lines 195-204: “Most of the enantioselective analyses on lactic, 2-hydroxybutanoic and 2-hydroxy-3-methylbutanoic acids, so far, did not report on significant *ee*-s of these molecules in meteorites, which contrasts with their amino acid analogues alanine and valine with reported %*ee*_L of up to 33.0% and 60.3% (Tab. 1), respectively. However, it should be noted that the number of reports on meteoritic amino acids is notably higher compared to HCAs, the significantly large %*ee*_L values in amino acids are often critically discussed on the basis of potential terrestrial contamination, and recent studies^{21,67} have found meteoritic amino acids in racemic ratios as well. In addition, the magnitudes of *ee*_L induced in the aliphatic chain HCAs by stellar CPL are expected to be lower than %*ee*_L uncertainties of enantioselective meteoritic analyses reported to date.”

REVIEWERS' COMMENTS:

Reviewer #1 (Remarks to the Author):

Thank you for carefully incorporating the suggestions. Please also change the text on the newly added line 215 from "...errors of around $\pm 5.0\%$..." to "...errors of around $\pm 5\%$ " as there is no need for two significant figures for this approximation.

Reviewer #3 (Remarks to the Author):

The manuscript has been modified based on the reviewers' comments including mine. So, I recommend accepting this manuscript for publication.